# Differences in Donor Animal Production Stage Affect Repeatability of In Vitro Rumen Fermentation Kinetics

**DOI:** 10.3390/ani13182993

**Published:** 2023-09-21

**Authors:** Britt Jantzen, Hanne Helene Hansen

**Affiliations:** Department of Veterinary and Animal Sciences, University of Copenhagen, Grønnegårdsvej 3, 1870 Frederiksberg C, Denmark; hhh@sund.ku.dk

**Keywords:** fermentation kinetics, IVGPT, repeatability, rumen fluid

## Abstract

**Simple Summary:**

A simple screening technique for nutritive value of feed and feed additives is used in many countries. This technique measures the gas produced from feed and additives as they ferment in rumen fluid. The technique was proven previously to yield repeatable results within laboratory and was reproducible across laboratories when the same feed was fermented in fluid from different animal species in different countries with different basal diets and feeding conditions. To our knowledge, there are no published results of repeated fermentations in the same lab using the same species of animals for donating the rumen fluid, but using animals in different production stages. The present research investigated the repeatability of results from 17 fermentations using the same feed. The fermentations were undertaken either using rumen fluid from continuously fed lactating cows or heifers that were fasted for 12 h before fluid collection. There were significant differences between the fermentation results when using either rumen fluid from lactating cows or heifers for pH, and gas production before 24 h, suggesting that donor animal production stage may be more important than animal species.

**Abstract:**

In vitro gas production techniques (IVGPT) are widely used to screen feeds and feed additives to reduce the number of animals needed for experiments, which in turn, reduces costs and increases animal welfare. However, information about repeatability is scarce. The objective of this study was to evaluate the variation from in vitro gas production fermentations in the same laboratory using the same feed substrate. The source of rumen fluid used in the fermentations was from two different farms with either cannulated lactating dairy cows or cannulated fasting heifers, representing two distinct stages of production (donor types). Seventeen 24 h fermentations, undertaken during a year, were used to evaluate the variation between the following parameters: gas curve parameters, baseline-corrected total gas production (TGP (mL at Standard Temperature and Pressure (STP))/g incubated dry matter (DM)), methane concentration (%) and yield (mL gas at STP/g DM), pH and degraded dry matter (dDM). Significant differences between donor types were found for the pH of the rumen fluid from individual animals and pH of fermented fluid. However, no significant differences were observed within donor type. The means for methane concentration and yield, after 24 h of fermentation, were not significantly different between or within donor types. Rate of early gas production was significantly different between donor types, but baseline-corrected TGP was not significantly different at 24 h. No dDM differences after 24 h of fermentation between or within donor types were detected. Gas production curves were different between donor types, being either a monophasic version of the sigmoidal model or an exponential curve for the heifers and the production animals, respectively. No differences were observed within type. Repeatability of rumen fluid (CV_RF_), calculated as the coefficient of variation, and the associated parameters, which were investigated, was best for methane yield (CV_RFALL_ = 0.3%) and least for TGP at 3 h (CV_RFALL_ = 3%). Repeatability was dependent on donor type.

## 1. Introduction

The human population is continuously expanding, which increases the demand for meat and milk from the cattle industry. FAO estimated an increase of 73% and 58% for meat and milk, respectively, from 2010 to 2050, which means an increase in greenhouse gas emissions [1]. This was supported by IFCN, who forecasted that milk production will increase by 35% from 2017 to 2030 [2]. Gerber et al. reported that livestock production contributes to approximately 14.5% of the total anthropogenic greenhouse gas emission, and enteric fermentation contributes to 39.1% of the total emission from this sector [3]. Another report by Opio et al. asserts this notion that, within the livestock sector, dairy cattle account for roughly 30% of the methane (CH_4_) emission [4]. An amplification of the cattle production is therefore of concern. To ensure that ruminant CH_4_ emission is minimized, a high-quality feed, which ensures meeting nutritional requirements, is crucial [5]. Conserved forages provide stable nutrition and are therefore widely used in countries with restricted growing seasons, such as Denmark. Maize silage (MS) is used not only in the Nordic countries but also used for all ruminant species worldwide [6,7]. MS is, therefore, an important component for ruminant diets to ensure nutritional fodder; hence, this feed has been used as one of the basal feed samples for all fermentations conducted in Denmark in the last decade. To ensure that MS and other fodder meet animal nutritional requirements, feed evaluation is necessary. 

In vitro fermentation techniques have been widely used to screen the nutritive value of feeds and assess their CH_4_ emission potential. Measurements based on these techniques together with a standard chemical composition analysis of the feed offers a rapid and cost-effective alternative to in vivo determination of nutrients [8,9]. In addition, these techniques also reduce the number of experimental animals used, especially when testing a large number of different additives, carriers, media, and feeds, which can, in turn increase animal welfare [8]. Depending on the study, these methods can be an effective option for investigating the CH_4_-reducing property of additives, before further evaluation in vivo. However, a positive effect found from treatment does not guarantee the same positive outcome in vivo or in production, and many factors may influence the results obtained [10]. A previous study showed that baseline-corrected and fitted IVGP curves were reproducible for hay and straw, despite differences in donor animal species (sheep and cows), from four laboratories in Europe using the same protocol [11]. Three of these laboratories used rumen fluid from non-producing cows (heifers or dry cows), while one laboratory collected rumen fluid from Segureña (a meat breed) wethers. Within the laboratory, variation was limited in this research as each lab used only one type of donor animal. Differences between the production state of the donor animals appear to be an important factor, and this difference may be critical to the results. 

Therefore, the objective of this study was to investigate the repeatability of fermentation kinetics using the same substrate in the same laboratory but with two sources of rumen fluid from cows in different production stages (donor types). We hypothesized that all fermentation parameters measured, with the same basal substrate, will not differ by donor types, nor between fermentation trials when corrected for baseline gas from a donor animal type.

## 2. Materials and Methods

### 2.1. Donor Animals

Seventeen 24 h fermentations were undertaken during a year, and in each of these fermentations, an internal standard with MS and samples with only rumen fluid were included. These samples were used to evaluate the variation between the following parameters: gas curve shape, baseline-corrected total gas production (TGP (mL at Standard Temperature and Pressure (STP))/g incubated dry matter (DM)), CH_4_ concentration (%) and yield (mL gas at STP/g DM), pH, and degraded dry matter (dDM). Twelve fermentations were undertaken with rumen fluid from two non-fasting cannulated lactating Danish Red dairy cows (mean lactation yield; 20 L energy-corrected milk (ECM), 2nd and 3rd parity) at Assendrup Hovedgaard (RF_A_). The last five fermentations were undertaken with rumen fluid from two fasting (12 h) cannulated Jersey heifers (age 6 and 8) at the University of Copenhagen Large Animal Hospital (RF_T_). The heifers were fed ad libitum haylage (with 90.5% DM, 3.78 MJ/kg DM and 4.9% protein/kg DM), for more than 6 weeks before the fermentation trials. The cannulated animals used were authorized according to Danish law (license number: 2012-15-2934-00648). The lactating cows were fed a Total Mixed Ration (TMR) of NaOH-treated wheat, rapeseed, MS, grass silage, wheat straw, and rapeseed cake (6.15 MJ/kg DM), for more than 6 weeks before the fermentation trials. The cannulated animals used were authorized according to Danish law (license number: 2018-15-0201-01462). The differences between the donor types investigated were primarily based on production stage but included the following: lactating cow vs. heifer, non-fasting vs. fasting before sampling, dry matter intake (DMI), and diet (due to stage of production). 

### 2.2. Experimental Procedures

Before collecting rumen fluid, a buffered medium was prepared according to the in vitro gas production protocol by Menke and Steingass [12]. Deionized water, macromineral solution, micro-mineral solution, and buffer were mixed and flushed with CO_2_ for 2 h before the addition of the rumen fluid to ensure anaerobic conditions. The temperature of the buffered media was maintained at 39 °C. A reducing agent containing sodium hydroxide and sodium sulfide was added 15 min before adding the rumen fluid. Rumen fluid was collected by adding rumen fluid and particulates from the rumen of each cow to preheated thermoses, and thereafter transported to the laboratory. Upon arrival, the content from each thermos was strained through a double layer of warm, wet commercial cheesecloth (roughly 5000 µm at dry). The substrate remaining in the cheese cloth was gently squeezed to ensure detachment of microorganisms from solid particles to be included in the inoculum. The pH from the rumen fluid of each cow was measured in the strained liquid. Equal amounts of fluid from each cow were used and mixed with the buffered rumen medium in a 2:1 ratio. The pH was measured in the buffered media with rumen fluid before and after dosing the bottles. The MS used for all of the fermentation trials were collected in 2019 and freeze-dried (Hetosicc CD 8, Heto Lab Equipment A/S, Allerød, Denmark) at −20 °C for 24 h at 0.1000 mbar with a final drying pressure at 0.0010 mbar. The dry material was ground with a laboratory mill (CT Cyclotex TM 193 TM, FOSS, Hillerød, Denmark) using a 2 mm sieve and stored until use. The DM content was determined by drying the material in a chamber (Binder GmbH, Bohemia, NY, USA) at 100 °C for 12 h, cooling in a desiccator until room temperature, and weighing. Crude protein (CP) was determined with Kjeldahl nitrogen content using the VELP Kjeldahl system (VELP Scientifica, New York City, NY, USA). Fiber was determined using the principles of plant cell wall fractionation proposed by Van Soest et al. [13]. Neutral detergent fiber with alpha amylase and without sulfite (aNDF) and acid detergent fiber (ADF) were determined using the protocol for the ANKOM Fiber Analyzer 200 (ANKOM, Rochester, NY, USA) [14]. Acid detergent lignin (ADL) was determined with the sulfuric acid method in a Daisy incubator (ANKOM, Rochester, NY, USA) according to the Daisy incubator lignin protocol [15]. The ash content was determined by burning the samples at 525 °C in a rapid heating chamber furnace (Carbolite RWF 1100, Carbolite Gero Ltd., Hope Valley, UK) for 16 h and weighing the dried and burned samples after cooling to the ambient temperature in a desiccator. The chemical composition of the MS is presented in Table 1.

A sample of 500 mg MS (±10 mg) was added to a 100 mL Duran bottle, except for bottles which contained only buffered rumen fluid. These “blanks” were used to establish the baseline minimum microbial activity present in the rumen fluid.

As soon as the rumen buffer media was mixed, 90 mL of buffered rumen fluid was added to all bottles, the headspace was flushed with N_2_ to remove CO_2_ and air, and the bottles were closed with the ANKOM^RF^ module head. 

Each fermentation unit consists of a 100 mL bottle and an ANKOM^RF^ (ANKOM Technology, Macedon, NY, USA) module head, equipped with a pressure sensor (pressure range: −10 to +4996 psi; resolution: 3.34 psi; accuracy ± 0.1% of measured values) including a microchip and a radio sender. During the incubations, the pressure changes in the headspace of the bottles, measured as a difference with respect to concurrently measured atmospheric pressure, were transmitted via a radio frequency to a PC at intervals of 1 min (live time). Gas accumulating in the headspace of the bottles was automatically released when the pressure inside the units reached 0.75 psi above ambient pressure. The cumulative pressure (psi) for each sample was calculated with the ANKOM^RF^ program. Absolute and cumulative pressures were recorded every 10 min.

A gas-tight sample bag (AL, CEK-1, GL Sciences, Eindhoven, Germany) was secured to the vent valve tube of the modules, to collect the produced gas. The modules were incubated at 39.5 °C in a thermoshaker (Gerhardt Analytical Systems, Königswinter, Germany) with 40 rotations per minute for 24 h.

After 24 h, the incubator was turned off, and all gas-tight sample bags were closed and detached from their modules and taken for CH_4_ analyses. The bottles were capped and placed in ice to stop fermentation. The content from each bottle was filtered through a pre-weighed filter bag with a porosity of 25 µm (F57, ANKOM Technology, Macedon, NY, USA) at the end of each fermentation trial to collect the undegraded residue, and the pH of the filtrate was measured. The filter bags were air dried at room temperature for 24 h and thereafter dried at 100 °C for 2 h, cooled to room temperature, and weighed to determine feed degradation.

The CH_4_ content in the gas-tight bags was measured with a gas chromatograph (GC) (Agilent 7820A GC, Agilent Technologies, Santa Clara, CA, USA) directly after the end of fermentation. The GC is equipped with a HPPLOT Q column (30 m × 0.53 mm × 40 µmm), which uses H_2_ as the carrier. Column flow was 5 mL/min, and the TCD detector was set at 250 °C with a reference flow of 10 mL/min. From each gas-tight bag, a gas sample of 250 µL was taken and manually injected into the GC. This was conducted at least twice for each bag to achieve analytical replicates with <10% deviation, and the average of the analytical replicates used for analyses. Run time was 3 min at an isothermal oven temperature of 50 °C. Calibration curves were calculated for each fermentation trial from standards containing 1%, 2.5%, 5%, 10%, 15%, and 25% CH_4_ in nitrogen (Mikrolab A/S, Aarhus, Denmark). The produced total CH_4_ volume and yield were thereafter calculated.

### 2.3. Calculations and Statisical Analyses 

For each fermentation, a modified procedure, as recommended in the protocol for the ANKOM^RF^ Gas Production system (ANKOM, Rochester, NY, USA) and Menke and Steingass, was used [12,16]. The gas pressure produced from the blank bottles was subtracted from the pressure production from bottles with substrate. This was conducted with all values until gas was absorbed from the headspace into the rumen fluid, causing a diminishing rate of production. Thereafter, the maximum gas pressure from the blank bottles was subtracted from all sample substrate values. Accumulated, blank-corrected gas pressure (psi) was converted to mL of gas STP/g incubated DM by using the ideal gas law.
V=nRT/P

The yield of gas for each bottle was calculated from the volume of gas (V).
mL gas/g DM = V/g DM in the sample              (corrected for baseline gas pressure)

The maximum gas production is reported as the accumulated gas production at the end of fermentation, half of this value is reported as “H” and the time when this occurs is reported as “H1”. The slope between each 10 min measurement is the rate of production, and the maximum rate is reported as mL gas/g DM per hour (Vmax) and the time at which this occurs is reported as “Tmax”.

The biomass filtered from the blank bottles is considered as minimum microbial biomass and is subtracted from the undegraded residues to determine dDM.

DM degradation was calculated as
dDM = 1 − ((Dry weight of the bag after fermentation − empty bag weight)/sample DM) (corrected for baseline microbial biomass)

TGP from individual bottles were considered technical replicates and were not included in the average if the deviation after 12 h was >10% [10]. 

The gas chromatography standard curves allows for the calculation of prediction equations (R^2^ > 0.985) for the relationship between area under the curve (total gas injected) and percentage of CH_4_. This allows the percentage of CH_4_ in each bag to be calculated. The total yield of CH_4_ was calculated from the total gas production and concentration of CH_4_ measured in the collected gas.

A full linear regression model was used for all parameters, to determine significant interactions between fermentation trial and donor type using R in the NLME package [17,18]:*Yij* = *μ* + *αi* + *βi* + (*αβ*)*ij* + ε*ij*
where *Yij* is the value for the response variable (dDM, pH, TGP at chosen times, CH_4_ concentration, and yield) in the fermentation trial *i* using the rumen fluid from donor animal *j* (T or A); *μ* is the overall mean, *α* is the fermentation trial (fixed) effect, *β* is the effect of the donor type, (*αβ*) is the interaction effect between trial and donor type, and ε is the error term. Model reduction was undertaken by removing non-significant variables. 

If the interaction was significant, the data was split by donor types and re-tested, and repeatability was calculated for each donor type. *t*-tests with the Welch approximation were undertaken to determine differences in the unbalanced number of fermentations. The repeatability was calculated for all fermentation trials using an unbiased estimate of the standard variation for trials and donor types, and thereafter, the coefficient of variation for rumen fluid was calculated as follows: $$\sigma =\sqrt{\sum_{i}^{n}{\left(X_{i}-X\right)}^{2}/(n-1)},$$
where σ is the unbiased estimate of the standard deviation for fermentation trials and/or donor type, (X_1_, X_2_, X_3_,…, X_n_) are the fermentations, and X is the mean of all trials and/or donor type.
CV_RF_ = (σ/X)100,
where CV_RF_ is the coefficient of variation for repeatability of rumen fluid, σ is the unbiased estimate of variation for trials and/or donor types, and X is the overall mean of the biological replicates. The abbreviations CV_RFT_, CV_RFA_, and CV_RFALL_ designate the coefficients of variation for fermentation trials with rumen fluid from fasted heifers (CV_RFT_), production cows (CV_RFA_), and for all fermentation trials (CV_RFALL_). 

## 3. Results

The chemical composition of the MS used as a basal feed substrate in all fermentation trials can be observed in Table 1.

The average dDM for RF_T_ and RF_A_ was 60.9% and 58.7%, respectively (Table 2). These values were not significantly different (*p* > 0.05). There was also no significant difference within the donor type (*p* > 0.05). 

The average pH values of the rumen fluid from individual cows, before the addition of buffer media, were significantly different (*p* < 0.05), with an average of 7.00 and 5.9 for the RF_T_ and the RF_A_, respectively (Table 2). However, there was no significant difference in pH within donor type over all of the fermentations. The pH of the individual bottle filtrate between donor types, after fermentation, was also different (*p* < 0.05). The average rumen fluid pH values after fermentation were 6.85 and 6.71 for the RF_T_ and the RF_A_, respectively. 

The TGP at time points 3, 6, 9, 12, and 24 h can be observed in Table 3. At 3, 6, 9, and 12 h, there were significant differences (*p* < 0.05) between the two donor types, but no significant difference (*p* > 0.05) within donor type during the fermentations was observed. After 24 h, there were no significant differences (*p* > 0.05) between or within donor types. Figure 1 shows the average TGP for all fermentation trials for Taastrup and Assendrup. 

There were large differences for all curve parameters by donor type (Table 4). There was a significant 9 h difference between the time of Vmax between the donor types but only a 2.5 mL difference in the average maximum rate of gas production. This suggests that, despite differences in time when the fermentation parameters occurred, the total gas and rate of gas production were similar.

The results for CH_4_ concentration (% in collected gas after 24 h of fermentation) were 9.2% and 9.9% for Taastrup and Assendrup. The yield of CH_4_ was 15.9 for RF_T_ and 17.9 for RF_A_(Table 5). There was no significant difference (*p* > 0.05) between or within donor types for both parameters.

## 4. Discussion

The IVGP system can be used to evaluate feeds and a broad spectrum of additives, before a possible in vivo fermentation. This system is advantageous because it is less expensive than in vivo or respiration chamber techniques and reduces the number of animals used, but sampling techniques, inoculum preparation, and donor animal management can be sources of variation and therefore have a substantial cumulative effect on the in vitro fermentation [10]. To ensure consistent results that can permit comparison between studies, repeatability should be evaluated, which was the objective of this study. Measures of biological replication in the same laboratory using the same protocol were calculated as coefficients of variation as conducted by Cornou et al. [11]. 

Evaluation of the variation was conducted on the following parameters: baseline-corrected TGP, gas curve parameters (time when half of the maximum gas was produced, maximum rate of gas production, and time of maximum rate of gas production), CH_4_ concentration and yield, pH, and dDM.

The results for the blank-corrected total gas production showed significant differences between donor types before 24 h and no significant difference at 24 h. Blank correction is conducted to separate the gas production of the baseline microbial activity of the original rumen fluid from the samples being tested. The baseline microbial activity varied extremely in a ring test of IVGPT from four European laboratories testing the same substrate without blank correction and with different donor animals [11]. Their gas production results ended up to be similar when incorporating the principle of blank correction for the baseline microbial activity. Menke and Steingass also suggest blank correction to minimize the variation in gas production [12]. However, none of the animals in the previous tests were high-yielding production animals. Our TGP results correspond to findings of Cone et al., who showed that the gas production rate for rumen samples taken from sheep and cattle was different until 24 h [19]. Calabro et al. compared rumen fluids taken from buffalo and sheep and observed a higher production rate and extent of degradation when using rumen fluid from sheep, but these differences in fermentation kinetics depended on the substrate used [20]. Fiber-rich substrates such as hay and straw yielded bigger differences, whereas the differences were negligible when using barley grain. Muetzel et al. compared rumen fluid from dry cattle (Holstein X Jersey) and wethers (no breed mentioned) and found that the TGP, when using ryegrass hay as a substrate, was unaffected by donor animal species [21]. Our results showed no significant difference within donor type at 6, 9, 12, and 24 h, but a significant difference was found in the third hour of fermentation when using rumen fluid from Taastrup. This was due to a 100% TGP variation during the start of the fermentation. Significant differences between donor types at 3, 6, 9, and 12 h were observed. The maximum coefficient of variation within each and overall donor types at the chosen times of fermentation was 12% (fermentations at three hours using Taastrup rumen fluid). Early fermentation kinetics have also been shown to vary greatly, based on the age of animals [22]. 

Collection procedures at Taastrup concur completely with the recommendations from Martínez et al. [23]. They proposed collecting rumen fluid samples before morning feeding to minimize the effect of the diet composition on the rumen microbiota and metabolites. Menke and Steingass also found that sampling before feeding lowered the variation in activity and composition in the inoculum, which then minimized the donor animal diet influence on the TGP [12]. Huntington and Givens compared gas production profiles from cows fed either a grass silage/barley grain diet (80:20) or a barley straw diet [24]. Their results found no differences in the gas production profile when sampling prior to morning feeding. However, Payne et al. observed that the variation between technical replicate bottles of TGP was less when the inoculum was collected either 4 or 8 h post-feeding, compared to 2 h before feeding [25]. The lactating cows at Assendrup were not subjected to fasting, as TMR was available at all times. However, rumen fluid collection was conducted early morning for practicality and to coincide, as best possible, with the time of collection at Taastrup. Cone et al. observed that rumen sampling time did not affect TGP, but the rate of fermentation was different depending on the lapse of time since the last feeding [26]. This is due to diurnal changes in the rumen microbiome, both in terms of metabolic activity and abundance, which is dependent on the level of feed intake and diet [27,28]. These contradictory findings substantiate the importance of keeping sampling procedures as identical as possible, when a series of studies are conducted over time. Another factor influencing the microbes is the transport time from the farm to the laboratory. From Assendrup to the laboratory, the distance is longer, which can make the transport time from Taastrup to the laboratory faster, and transport time can vary by up to 35 min. This difference could be a source of some of the variations observed in the TGP [19,29].

The dDM showed no differences between or within donor types after 24 h of fermentation, which can also be observed in the gas production curves. If the fermentation duration was reduced to 16 h, the time at which TGP differences were observed, significant differences in the dDM might have been observed. Differences before 16 h should be considered as passage rate can be as low as 6–9 h for soluble nutrients and concentrates [30].

The results for the pH show a significant difference between the donor types, but no difference within donor types over the course of the fermentation trials. The pH was 5.9 for the rumen fluid from the concentrate-fed cows in Assendrup and 7.0 for the forage-fed heifers from Taastrup before fermentation. When added to the same buffer media and fermented with the same feed substrate, the pH after 24 h of fermentation still showed significant differences, suggesting this is a major determinant for fermentation parameters when using IVGPT. These differences concur with the knowledge about the rumen digesta phases (solid vs. liquid) and particle size. Normal cow ruminal pH varies from 5.5–7, depending on the diet composition [31]. Different diets promote growth of different microorganism and their metabolites, and their composition and nutrient availability are the largest factors affecting microbial growth in the rumen and, therefore, the microbial activity of the inoculum [28,31]. A study conducted by Latham et al., on non-lactating cows, found that substituting an all-hay diet with wither 800 g/kg barley or maize grain decreased the cellulolytic bacterial count from 10^7^ to 10^6^ (hay to barley) and from 10^7^ to 10^3^ with maize grain and lowered the pH [32]. The level of feed intake of the donor animal is also important to consider since a greater DMI decreases the retention time in the rumen, which decreases the time available for microbial feed degradation [33]. Rumen pH and cellulolytic and proteolytic activity are also thought to be influenced by the DMI level, which in turn influences the growth rate of the microbes and the metabolic potential of the inoculum. To decrease the diurnal variation in the rumen inoculum, an increased feeding frequency can be implemented. Le Liboux and Peyraud observed a decrease in the post-feeding variation for the rumen pH when they increased the feeding frequency from two to six times per day for lactating cows with a milk yield between 24.3–25.9 kg per day [34]. A fasting duration of 12 h for the animals at Taastrup compared to TMR available ad libitum at Assendrup is most likely to cause differences in the rumen fluid, which can be observed in the pH parameters even after fermentation of the same substrate in standard buffer solution.

Another factor influencing the pH level before fermentation is the sampling site. A study conducted by Pei et al. showed a difference in the abundance and diversity of the microorganisms in the rumen depending on the collection site (solid material vs. liquid phase) [35]. Storm and Kristensen found that the pH differed 0.4–0.6 pH units depending on where in the rumen the sample was taken from, which corresponds to findings by Shen et al., who found that rumen fermentation parameters differ between sampling collection sites [36,37]. While the protocol instructions state sampling from defined areas in the rumen, authorized technical assistance is required for rumen sampling in Denmark. This creates variation, as different people collect the fluids. The pH before fermentation was 18.6% greater for the fiber-fed animals than the concentrate-fed cows, while it was only 2% greater after fermentation in the same buffer solution. Our buffered rumen media had a bicarbonate concentration of 73.8 mM, which is less than the critical level found in a study by Patra and Yu [38]. They found that bicarbonate concentrations in the buffer solution above 80 mM should be avoided to minimize non-microbial CO_2_ production associated with pH changes. This suggests that the buffer solution plays a crucial role in ensuring minimal variation during fermentation, allowing the fermentation results to reflect the substrate being tested. 

CH_4_ concentration and yield showed no significant differences between or within donor types. Enteric CH_4_ production (concentration and yield) is influenced by many factors such as the DMI level, diet, lactation stage, and genetics amongst others. DMI is a major determinant for both rumen pH and enteric CH_4_ [39]. A positive correlation between DMI and CH_4_ in high production (Holstein × Friesian) dairy cattle was also found by Dijkstra et al. [40]. Increasing the DMI of a good quality dairy feed will generally result in an increased amount of organic matter fermented in the rumen with the associated increased TGP and CH_4_. Another factor that also influences the CH_4_ production from the rumen is the dietary composition of the rations (forage- vs. concentrate-based diet). A higher inclusion of forage in a diet is expected to increase the cellulolytic activity on the rumen microbiota, which then increases the CH_4_ concentration [41]. The ratio of CH_4_ yield/DMI has been shown to increase during lactation and the CH_4_ production increases according to parity, which can be explained by the fact that milk production and subsequently DMI increases accordingly [42,43]. Differences in the genetic makeup between and within breeds can also play a role in the amount of CH_4_ emitted by the cows [44,45]. Difford et al. also determined that the donor rumen microbiome composition influenced enteric CH_4_ production, measured from the cow. However, all of these factors should not influence IVGP parameters (TGP, CH_4_ concentration, and yield), when baseline microbial activity has been taken into account and fermentation duration is 24 h [45]. 

The repeatability, calculated as CV_RFT_ and CV_RFA_, varied from 3–12% and 1–3%, respectively. The CV_RFAll_ was less than 1.5% for dDM, pH before and after fermentation, CH_4_ concentration, and yield. This indicates the usefulness of the IVGP techniques but emphasizes the necessity of standardized protocols. This type of repeatability analysis should be undertaken at laboratories where IVGPT is routinely run to evaluate feeds and additives. 

Our results lead to a rejection of the original hypothesis that all parameters measured with the same basal substrate will not differ by donor animals. The repeatability within donor types over fermentation trials was less than between donor types and not significantly different. These results indicate that donor animal production state and therefore DMI are important variables that influence IVGP results, despite the global application of the technique and lack of differences in previous studies. In addition, IVGP results might not reflect in in vivo fermentation kinetics, and if our fermentation had been reduced to 16 h, the CH_4_ concentration and yield might mirror true kinetics better.

## 5. Conclusions

The objective of this study was to investigate the repeatability of fermentation kinetics using the IVPGT with blank correction from rumen fluid donors in different production stages. Blank correction was not sufficient to account for kinetic parameter differences. Within donor types, the repeatability of IVGP trials is high. The variation over fermentation trials within donor type is not significantly different for all parameters if a strictly standardized protocol is used. The variation between donor types will be significantly different for the following parameters: pH before and after fermentation and the total gas production until 24 h (if high yielding, lactating cows are compared to heifers). Significant differences will primarily reflect the dietary composition and DMI of the donor. This type of evaluation should be performed by laboratories routinely conducting IVGPT for feed and additive evaluation.

## Figures and Tables

**Figure 1 animals-13-02993-f001:**
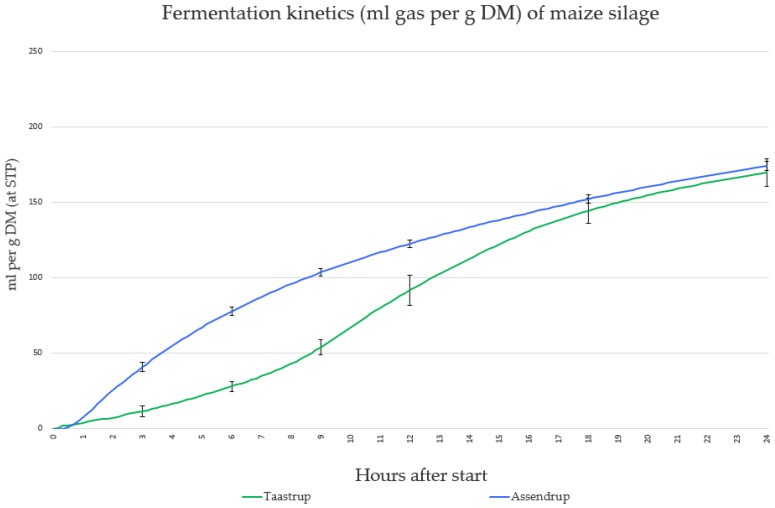
The average total gas production curve for all fermentation trials conducted with either fasting heifers from Taastrup or lactating cows from Assendrup with the standard error of the mean error bars.

**Table 1 animals-13-02993-t001:** Chemical composition of maize silage (MS) used in the study.

Item	Maize Silage
Dry matter (DM) %	92.6
Crude protein (%)	9.0
aNDF% (DM)	39.4
ADF% (DM)	21.1
ADL% (DM)	1.2
Ash% (DM)	3.3

aNDF: neutral detergent fiber with amylase. ADF: acid detergent fiber. ADL: acid detergent lignin, including acid insoluble ash. Ash: residue after burning in a rapid heating chamber furnace at 525 °C for 16 h.

**Table 2 animals-13-02993-t002:** Degraded dry matter (dDM%), pH of rumen fluid before and after 24 h of fermentation of MS from lactating cows and fasting heifers, and repeatability of these parameters.

Variable/Rumen Fluid Source	RF_T_	SEM	CV_RFT_ (%)	RF_A_	SEM	CV_RFA_ (%)	CV_RFAll_ (%)
N	5			12			17
dDM (%)	60.9	0.01	0.2	58.7	0.01	0.5	0.2
pH before addition to media	7.00 _a_	0.03	0.3	5.90 _b_	0.06	0.2	0.3
pH after fermentation	6.85 _a_	0.01	0	6.71 _b_	0.01	0	0

_a,b_ Values within a row are different if subscripts differ (*p* < 0.05). N: the number of fermentation trials. dDM: degraded dry matter. RF_T_ and RF_A_ are rumen fluids from fasting heifers from Taastrup (T) or lactating cows from Assendrup (A), respectively. CV_RFT_, CV_RFA_, and CV_RFAll_ are coefficients of variation for rumen fluid obtained from Taastrup (T), Assendrup (A), or both (All), respectively. SEM: standard error of the mean.

**Table 3 animals-13-02993-t003:** Total gas production (TGP: mL gas at Standard Temperature and Pressure (STP)/g incubated DM) and repeatability at chosen time points during fermentation of maize silage from lactating cows and fasting heifers.

Time/Variable	TGP_RFT_	SEM	CV_RFT_ (%)	TGP_RFA_	SEM	CV_RFA_ (%)	CV_RFALL_ (%)
N	5			12			17
3 h	11.41 _b_	3.47	12.2	40.80 _a_	3.55	2.6	3.1
6 h	27.93 _b_	3.27	6.5	77.59 _a_	3.31	1.3	2.6
9 h	53.82 _b_	4.81	5.0	103.32 _a_	2.99	0.9	1.9
12 h	91.68 _b_	10.08	6.1	122.48 _a_	3.08	0.8	1.1
24 h	169.58	9.15	3.0	174.45	3.58	0.6	0.5

_a,b_ Values within a row are different if subscripts differ (*p* < 0.05). N: the number of fermentation trials. TGP_RFT_ and TGP_RFA_ are average total gas productions (mL at STP/g DM) from MS fermentations conducted with rumen fluid from fasting heifers from Taastrup (T) or lactating cows from Assendrup (A), respectively. CV_RFT_, CV_RFA_, and CV_RFAll_ are coefficients of variation for rumen fluid obtained from Taastrup (T), Assendrup (A), or both (All), respectively. SEM: standard error of the mean.

**Table 4 animals-13-02993-t004:** Curve parameters of all fermentations of maize silage using rumen fluid from lactating cows and fasting heifers.

	N	H	SEM	H1 (Hours)	SEM	Vmax	SEM	Tmax (Hours)	SEM
TGP_RFT_	5	84.8 _b_	4.6	9.7 _a_	1.58	16.1	1.43	11.4 _a_	0.83
TGP_RFA_	12	87.2 _a_	1.7	6.8 _b_	0.29	18.6	0.93	2.3 _b_	0.25

_a,b_ Values within a column are different if subscripts differ (*p* < 0.05). N: the number of fermentation trials. TGP_RFT_ and TGP_RFA_ are average total gas productions (mL at STP/g DM) from MS fermentations conducted with rumen fluid from fasting heifers from Taastrup (T) or lactating cows from Assendrup (A), respectively. H: the average of half of the maximum gas produced in all experiments. H1: time when half of the average maximum gas is produced (hours). Vmax: average maximum rate of gas production per hour. Tmax: average time when maximum rate of gas production occurred (hours). SEM: standard error of the mean.

**Table 5 animals-13-02993-t005:** Methane yield (mL gas at STP/g incubated DM) and concentration (%) in collected gas after 24 h of in vitro fermentation of maize silage using rumen fluid from lactating cows or fasting heifers.

	RF_T_	SEM	CV_RFT_ (%)	RF_A_	SEM	CV_RFA_ (%)	CV_RFALL_ (%)
N	5			12			17
Methane yield (mL at STP/g DM)	15.9 _a_	1.54	5.5	17.9 _a_	1.18	2.1	1.4
Methane concentration (%)	9.2 _a_	0.52	3.2	9.9 _a_	0.5	1.6	1.0

_a_ Values within a column are different if subscripts differ (*p* < 0.05). N: the number of fermentation trials. RF_T_ and RF_A_ are rumen fluids from fasting heifers from Taastrup (T) or lactating cows from Assendrup (A), respectively. CV_RFT_, CV_RFA_, and CV_RFAll_ are coefficients of variation for rumen fluid obtained from Taastrup (T), Assendrup (A), or both (All), respectively. SEM: standard error of the mean.

## Data Availability

Not applicable.

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
