# Peer review of "Differences in Donor Animal Production Stage Affect Repeatability of In Vitro Rumen Fermentation Kinetics"

_animals, 2023, doi:10.3390/ani13182993_

Round 1

Reviewer 1 Report

Title:

title is nondescript and repetitive

Experimental design

The work is not based on a predefined experimental design but uses data from a number of fermentation runs (17 incubations) to build a data set. For this reason, the experimental design is not balanced but still allows to respond to the experimental hypothesis of the research.

Objective

The objective of the work is “to investigate the repeatability of fermentation kinetics using rumen fluid from two types of donor animals with the same substrate in the same laboratory”.

However, as well described and argued by the Authors, the differences between the two experimental treatments are not limited to the donor animal but also to the feeding conditions, the place, etc. Therefore, the definition "donor animals" is not representative of the differences between the “treatments”. For this reason, I suggest reporting an alternative definition: "source of rumen fluid"? or “inoculum source”?

M&M

In some parts, the analytical methodologies are not accurately described, and the use of some not intuitive acronyms complicates the understanding of the text and tables (RFA e RFT, RFAll to define the different "donor animals") .

Line 73: Always report the meaning of acronyms as they first appear (dDM). (I prefer the acronym DMd (g/g) for “dry matter digestibility/degradability” and dDM for “degraded(digested) dry matter” (g))

Experimental design

The repeatability of measurements as well as the SEM values is strictly related to the number of replicates incubated in each run. This information is not reported in M&M and in tables and is crucial for evaluating the repeatability of the measurements obtained with the two types of rumen fluids. If the number of replicates chances between experiments, this should be also discussed. Similarly, the use of an unbalanced number of fermentation runs (5 vs 12 runs) affects the values ​​of SEM and CV. This should also be discussed.  

Line 75. The importance and role of the collection site of the rumen fluids (Assendrup and Taasrupt?) can only be understood after reading the discussion of the work. In fact, the two "donor animals" differ in multiple factors that should be emphasized in a table. I suggest reporting and defining here the acronyms RFa and RFt.

L152 The acronym STP is never defined in the text and, probably, it can be removed throughout the text.

L157 Please define this variable. TGP?

The “base line blank correction” procedure must be accurately described because such correction has considerable relevance to the results of the work.

159 “Baseline microbial biomass”? Please define and describe the analytical determination.

159 Please, describe the model used to calculate the kinetic parameters reported in table 4. These parameters are usually defined as: A (ml/g DM), T1/2 (h), Vmax (rate of gas production) and Tmax (h). I never see the parameter H (“half of maximum? gas production” à please report the parameter “A” as “asymptotic gas production”, ml/g DM).

L177-180. This procedure is not consistent with data reported in tables. I recommend always reporting both the repeatability of the two groups of animals and the overall repeatability.

L181 The formula to calculate S2 is not correct.

182 S2 is not a “standard deviation”, please revise the sentence.

184 if I correctly understand, the symbol “σ” should be replaced with S (the square root of the variance)!

 Table 2 check the SEM dDM for RFt

Table 2, 3, 4, please report consistent acronyms for the two “types of donor animals”. RFt … TGPt ????

Table 2, 3, 4, In general the letters a,b are reported when different, and not when they are the same,

Table 3 the number 17 is missing in the last column.

Table 4 why the acronym TGP instead RF? Please report data in the same arrangement of other tables.

Table 4 why the RFall is not reported? These parameters are significantly different for RFt e RFa

L269 As suggested by the Authors, the contribution of blanks on the GP is an important source of variation. The correction (or not) of the GP values ​​for blanks is a widely discussed aspect in the literature. For this reason, it would be interesting to analyse the variability (and the repeatability) of blank GP kinetics as well as the pH values at the end of the runs. In my opinion, blanks are not necessarily an indicator of the "basic microbial activity" because in the absence of the fermentation substrate, bacterial autolysis processes begin which lead to the release of ammonia, the increase of pH that can lead to the solubilization of CO2 gas in the medium as carbonic acid and, at the end, the decrease of GP after few hours from the beginning of incubation. For these reasons it is always advisable to report the values ​​of blanks especially in a methodological work.

 Conclusions

Consistent with the search results

Author Response

Thank you so much to the reviewer for your time and suggestions.

The attachment file is our reply to the comments. We have used “–“ to show the start of a comment from us.

Reviewer 2 Report

Article

animals-2564364

Repeatability of fermentation kinetics using in-vitro gas fermentation

GENERAL REMARKS

Dear authors,

I have evaluated the manuscript identified as animals-2564364. The manuscript adds, albeit only slightly in my opinion, a further piece to the sector literature, providing further insight into the limits and potential of in vitro fermentability through the automated gas production technique. So, the manuscript's contribution to extending literature knowledge is appreciated. I have not highlighted major problems in the manuscript, which is scientifically robust both in terms of methods and data analysis. However, even if patchy, there are several points for improvement to address. Furthermore, the manuscript completely lacks the simple abstract required by the standards of the journal. Therefore, a major revision is required.

Further specific comments are listed below, point by point.

SPECIFIC COMMENTS

L 7: please, add a Simple Summary of no more than 200 words in one paragraph that contains a clear statement of the problem addressed, the aims and objectives, pertinent results, conclusions from the study, and how they will be valuable to society. This should be written for a lay audience, i.e., no technical terms without explanations. No references are cited and no abbreviations. An example could be found at https://www.mdpi.com/2076-2615/6/6/40/htm. Thanks.

L 8: animals needed for what? I ask the authors to be more specific, thanks.

L 17-18: it is not clear what and/or between what the lack of differences refers to. I always assume at the pH, but I think it's more useful that this (or another) is specified, thanks.

L 31 (and through the text): as per journal standard all abbreviations must be specified on the first mention. These also include those referring to standard acronyms (for example, ECM, DIM, DM, BW, etc.) for animal production manuscripts. Unfortunately, the newspaper does not include a list of abbreviations. Thanks.

L 32: according to the journal template (https://www.mdpi.com/journal/animals/instructions) the Vancouver style should be adopted for the references. So, references must be numbered in order of appearance in the text (including table captions and figure legends) and listed individually at the end of the manuscript. In the text, reference numbers should be placed in square brackets [ ], and placed before the punctuation; for example [1], [1–3], or [1,3]. For embedded citations in the text with pagination, use both parentheses and brackets to indicate the reference number and page numbers; for example [5] (p. 10). or [6] (pp. 101–105). Authors are required to update the text, thanks.

L 39-45: in my opinion, this specific reference to the Danish reality gives a note of extreme localism. Corn silage represents the forage base of rations for several dairy animals (not only cows) in different agricultural areas of the world (almost all where there is availability of irrigation water). To underline the expressed concept and, at the same time, avoid a marked localism (in my opinion penalizing) the authors can also make a brief reference to further productive realities. For lactating buffalo farms, the authors can validly refer to https://doi.org/10.3390/agriculture12081219. Thanks.

L 51: in addition to the utilitarian aspects, in my opinion, the authors could also refer to aspects related to animals’ standard well-being in defining the advantages associated with the adoption of in vitro studies compared to in vivo ones. Thanks.

L 75: what diet was adopted for the non-fasting donor cows? For heifers, this has been specified. I ask you to update the text, thanks.

L 76: besides acronyms (see comment on line 31), I suggest defining ECM and parity values as mean ± standard deviation. Thanks.

L 77: if the donor cows were cannulated, the same was not declared for the donor heifers whose rumen fluid was used in the last 5 fermentations. I ask the authors, therefore, to specify how the rumen fluid was taken from the heifers, thanks.

L 78: how old are the heifers? This detail could be useful to hypothesize the levels of DMI on which the authors speculated in the conclusions.

L 99: please, delete the point after the “and” conjunction. Thanks.

L 102: the authors used a frozen substrate, an unusual condition compared to the most practical circumstances. This could lead to a reduction in the repeatability of the experimental results. I ask you to explain if this choice has a rationale and, above all if freezing does not involve changes in the fermentation attitude of the substrate. If freezing is a current practice or, better still, there are studies that point out the lack of interference, I ask you to indicate it in the text. Thanks.

L 106 (and in Table 1): what does the adjective "final" imply? I'm honestly not clear what is meant by "final DM". Thanks.

L 191 (Table 1): for none of the analytical parameters listed in the table, the analysis procedures are described in the materials and methods section. Although these are standard procedures, I ask you to update the text. Also, all acronyms must be explained in the table notes. Thanks.

L 147, 243, and somewhere else: please, add a space between "g" and "DM". Thanks.

L 194 (and through the text): according to the journal template, the p-value should be reported in lowercase and italics. Thanks.

L 202 (Table 2): The meaning of the "N" parameter is elusive and not specified in the notes. This also applies to tables 3 and 5. I ask you to review this aspect. Thanks.

L 223 (Figure 1): Figure 1 is formatted with different fonts. Please, format the table according to the font required by the journal. Thanks.

L 255: discussion is very effective. Well done!

L 413-416: I find these concluding statements somewhat speculative since there is a strong homogeneity between the two groups of donor animals (the cows and heifers are of the same breed within the groups) and no DMI measure supporting this statement has been presented in the text (although it is obvious that there are differences between cows and heifers). I suggest refining the conclusions strictly on the observed results, and strictly recalling the experimental hypothesis.

Author Response

(The authors gave the same response as above.)

Round 2

Reviewer 1 Report

The authors revised the text and accepted most of my suggestions. Now the text is much clearer. I believe that the work can be published in the present form

Reviewer 2 Report

Dear authors,
I reviewed the revised version of the animals-2564364 manuscript. Based on the changes made, I believe that the current version of the manuscript deserves to be published, in my opinion.
Congratulations